# Improved gridded ammonia emission inventory in China

Baojie Li[1], Lei Chen[1], Weishou Shen[1], Jianbing Jin[1], Teng Wang[2], Pinya Wang[1], Yang Yang[1], and Hong Liao[1]*

[1] Jiangsu Key Laboratory of Atmospheric Environment Monitoring and Pollution Control, Jiangsu Collaborative Innovation Center of Atmospheric Environment and Equipment Technology, School of Environmental Science and Engineering, Nanjing University of Information Science and Technology, Nanjing, 210044, China
[2] College of Oceanography, Hohai University, Nanjing, 210098, China

*Correspondence to*: Hong Liao (hongliao@nuist.edu.cn)

**Abstract.** As a major alkaline gas in the atmosphere, $NH_3$ significantly impacts atmospheric chemistry, ecological environment, and biodiversity. Gridded $NH_3$ emission inventories can significantly affect the accuracy of model concentrations and play a crucial role in the refinement of mitigation strategies. However, several uncertainties are still associated with existing $NH_3$ emission inventories in China. Therefore, in this study, we focused on improving fertilizer application-related $NH_3$ emission inventories. We comprehensively evaluated the dates and times of fertilizer application to the major crops that are cultivated in China, improved the spatial allocation methods for $NH_3$ emissions from croplands with different rice types, and established a gridded $NH_3$ emission inventory for mainland China with a resolution of 5 min × 5 min in 2016. The results showed that the atmospheric $NH_3$ emissions in mainland China amounted to 12.11 Tg, with livestock waste (44.8%) and fertilizer application (38.6%) being the two main $NH_3$ emission sources in China. Obvious spatial variability in $NH_3$ emissions was also identified, and high emissions were predominantly concentrated in North China. Further, $NH_3$ emissions tended to be high in summer and low in winter, and the ratio for the July–January period was 3.08. Furthermore, maize and rice fertilization in summer was primarily responsible for the increase in $NH_3$ emissions in China, and the evaluation of the spatial and temporal accuracy of the $NH_3$ emission inventory established in this study using the WRF-Chem and ground station- and satellite-based observations showed that it was more accurate than other inventories.

## 1 Introduction

Ammonia, a major form of reactive nitrogen, plays an important role in atmospheric chemistry, the ecological environment, and biodiversity (Sheppard et al., 2011; Zhang et al., 2018). As the major alkaline gas in the atmosphere, it can form $(NH_4)_2SO_4$ and $NH_4NO_3$ with $H_2SO_4$ and $HNO_3$ produced from the oxidation of $SO_2$ and $NO_x$, respectively, and contribute to the formation of secondary inorganic aerosols (SIA), thereby increasing the concentration of $PM_{2.5}$ (Fu et al., 2017). For example, in China, the contribution of agriculture-related $NH_3$ emissions to SIA and $PM_{2.5}$ is 29% and 16%, respectively (Han et al., 2020). This fine particle formation has led to substantial health and economic costs (Paulot and Jacob, 2014).

Haze pollution occurs frequently, particularly in eastern China, and is characterized by an extremely high concentration of $PM_{2.5}$, with a remarkably high proportion (20−60%) of SIAs (Ding et al., 2016; Elser et al., 2016; Wang et al., 2016). The Chinese government has taken effective measures to control $SO_2$ and $NO_X$ emissions, and a large number of studies have indicated that $SO_2$ and $NO_X$ concentrations and emissions have decreased in recent years (Zheng et al., 2018; Li et al., 2019b; Zhang et al., 2019). However, measures to reduce $NH_3$ emissions are limited. Satellite retrievals have shown an

increase in $NH_3$ vertical column densities (VCDs) in recent years (Warner et al., 2017; Chen et al., 2020), and China has become a global "hotspot" for $NH_3$ emissions and $NH_3$ pollution (Liu et al., 2013; Liu et al., 2019a). Such increases in $NH_3$ concentrations may reduce the effectiveness of particle pollution control achieved via $SO_2$ and $NO_X$ emission reduction (Wang et al., 2013; Fu et al., 2017). Therefore, to effectively control PM pollution and reduce SIA concentrations in China, strategies to reduce $NH_3$ emissions are urgently required.

Recently, $NH_3$ emission reduction has been proposed as a strategic option for mitigating haze pollution (Liu et al., 2019b). Several organizations and researchers have established gridded $NH_3$ emission inventories, such as the MEIC, PKU-$NH_3$, MASAGE_$NH_3$, EDGAR, and REAS (Streets et al., 2003; Paulot et al., 2014; Fu et al., 2015; Kang et al., 2016; Li et al., 2017a; Zhang et al., 2017; Zhang et al., 2018; Crippa et al., 2020; Kurokawa and Ohara, 2020). Based on these studies, substantial progress has been made in the development of $NH_3$ emission inventories. However, based on the results of previous

studies, $NH_3$ emissions in China have been estimated as 8.0–18.3 Tg/year (Zhang et al., 2017; Kong et al., 2019), which is indicative of large uncertainties. Unlike $SO_2$ and $NO_X$, which primarily originate from industrial plants, $NH_3$ mainly originates from agricultural activities, which are more difficult to evaluate. Of course, certain non-agricultural sources of $NH_3$ are also important. For example, vehicular emissions contain both $NH_3$ and $NO_x$ and may have a more effective pathway to particle formation, particularly in urban areas (Farren et al., 2020; Chang et al., 2016). The amount of $NH_3$ emitted by the transportation

sector has not been well quantified, and is generally thought to be underestimated (Meng et al., 2017; Farren et al., 2020). Such uncertainty limits the accuracy of $NH_3$ and PM concentration estimates simulated using atmospheric chemistry transport models. Fertilizer application and livestock waste were the largest contributors, accounting for more than 80% of total $NH_3$ emissions (Zhang et al., 2017; Kang et al., 2016); thus, the improvement of $NH_3$ emission inventories should primarily focus on these two sources (Zhao et al., 2020). To improve $NH_3$ emission inventory, several environmental factors, including wind

speed, temperature, and soil pH, have been considered in some studies (Paulot et al., 2014; Zhang et al., 2018; Zhao et al., 2020). The mass-flow approach, which considers nitrogen transformation at different stages of manure management to improve $NH_3$ emission inventories from livestock waste, has also been applied (Huang et al., 2012; Zhang et al., 2018).

    Monthly variations in $NH_3$ emissions are primarily caused by fertilizer application, according to previous studies (Huang et al., 2012; Zhang et al., 2018). However, fertilizer application-related $NH_3$ emission inventories need to be further

improved for the following reasons. First, in most existing studies, spatial differences in the quantity of fertilizer application in different provinces were considered but spatial variations with respect to the timing of fertilizer application were not, hence the uncertainty in the monthly rate of $NH_3$ emissions. For example, with respect to winter wheat, in North China, basal dressing is usually conducted from late September to mid-October, while in the Yangtze River Delta region, it is usually conducted

from late October to early November. Even in the same province, the difference in fertilizer application dates can be approximately 15 days. In most studies, fertilizer application dates were set to a specific month; however, this is not consistent with reality. Farmers in the same province usually apply fertilizers across months, rather than in a specific month. Additionally, the temporal and spatial differences in fertilization dates are critical to the accuracy of the NH$_3$ emission inventories. Second, there are significant differences in the spatial distribution of planted areas for different crops. This implies that different spatial proxies must be used for different crops. For example, in China, four rice types are cultivated, namely, early, late, single-season, and middle rice, which cannot be allocated using the same spatial allocation method.

Therefore, in this study, we established a 2016 NH$_3$ emission inventory for mainland China with a 5 min × 5 min resolution. To improve the accuracy of the emission inventory, we focused on improving the accuracy of NH$_3$ emissions from fertilizer application, and the latest methods in the literature were used for quantifying emissions from other sources, such as livestock waste. Finally, the inventory accuracy was evaluated using WRF-Chem and available ground station- and satellite-based observation data.

## 2 Methods and materials

This study was conducted in mainland China. Hong Kong, Macao, and Taiwan were excluded. Fifty emission sources, including fertilizer application, livestock waste, transportation, biomass burning, and agricultural soil, were considered. The categorization of these sources is presented in Table 1. The gridded NH$_3$ emissions ($E_{NH_3}$) were calculated according to Eq. (1):

$$E_{NH_3} = \sum_i \sum_j \sum_k A_{i,j,k} \times EF_{i,j,k}$$

(1)

where $i$, $j$, and $k$ represent the specific grid, source type, and month, respectively. $A$ represents the activity level (e.g., fertilizer application amounts corresponding to each crop, mileage of motor vehicles, etc.), and $EF$ represents the corresponding emission factor.

Reportedly, monthly variations in NH$_3$ emissions can be primarily attributed to differences in fertilizer application amounts (Huang et al., 2012; Zhang et al., 2018). However, fertilizer application-related NH$_3$ emission inventories have considerable uncertainties owing to several factors, including fertilizer type, crop type, fertilization times, fertilization dates, and environmental factors. Therefore, in this study, we focused on improving the accuracy of fertilizer application-related NH$_3$ emission inventories.

**Table 1. NH₃ emission sources in China.**

| Category | Subcategory | Category | Subcategory |
|---|---|---|---|
| Fertilizer application | urea | Residential & commercial | human excrement |
| | ammonium bicarbonate (ABC) | | indoor firewood combustion |
| | diammonium phosphate (DAP) | | indoor wheat burning |
| | NPK compound fertilizer (NPK) | | indoor rice burning |
| | other | | indoor maize burning |
| Livestock waste | beef cattle | | domestic coal combustion |
| | dairy cow | | domestic oil combustion |
| | goat | | domestic gas combustion |
| | sheep | Industry | synthetic ammonia |
| | rabbit | | nitrogen fertilizers production |
| | horse/donkey/mule | | wastewater treatment |
| | sow | | waste landfill |
| | fattening pig | | waste incineration |
| | camel | | coal combustion |
| | meat duck | | oil combustion |
| | meat goose | | gas combustion |
| | broilers | Other | agricultural soil |
| | laying hen | | nitrogen-fixing plants (soybean) |
| | laying duck | | nitrogen-fixing plants (peanuts) |
| Traffic | light-duty gasoline vehicles | | outdoor straw burning |
| | heavy-duty gasoline vehicles | | forest fires |
| | light-duty diesel vehicles | | grassland fires |
| | heavy-duty diesel vehicles | | |
| | motorcycles | | |

90

## 2.1 Improvement of fertilizer application-related NH₃ emission inventories

Five fertilizer types, including urea, ammonium bicarbonate (ABC), diammonium phosphate (DAP), and complex-fertilizer (NPK) were considered in this study. Additionally, several types of crops that are widely cultivated in China were also considered, including early, middle, late, and single-season rice; winter and spring wheat; spring and summer maize; cotton;

potato; spring and winter rapeseed; soybean; spring and summer groundnut; sugarcane; sugar beet; tobacco; apple; citruses; pear; and vegetables. Specifically, the following improvements were made to inventories of NH₃ emissions from fertilizer applications:

(1) In most previous studies, fertilization dates were set such that they did not change in space, but were fixed to a specific month; this is not consistent with the actual situation. Therefore, in this study, we comprehensively evaluated the fertilizer application timing and frequency for rice, maize, and wheat crops (these three plants constituted a total of eight sub-categories as shown in Fig. 1) in different regions by collecting data from a large number of studies online (primarily comprising reports on crop phenology in each province in 2016) and the technical guidelines for field management in each province (see a more detailed explanation in section S1). The respective amounts of each N fertilizer, applied in different months to each crop category, can be calculated using the following formula:

$$A_{i,j} = A_{total} \times P_j \times x_{i,j} \qquad (2)$$

where i indicates the month. $A_{total}$ indicates the total amount of each N fertilizer (ABC, DAP, NPK, urea, and other) applied to the different crop types in each province in 2016, which were calculated as a product of the planted cropland area and the fertilizer application rate per unit area of cropland, based on data from MARA (2017) and NDRC (2017). Over the whole growth period, wheat, maize, and rice generally need three applications of N fertilizer ($j$, namely the basal dressing, first topdressing, and second topdressing). $P_j$ represents the proportion of the total annual fertilization amount applied in the $j$-th dressing. The variation in $P_j$ across the different regions was also considered, based on farmer survey data (Wang et al., 2008) and other studies in the literature (Zhang et al., 2009; Zhang and Zhang, 2012). $x_{i,j}$ represents the probability of the $j$-th dressing in month $i$, and is calculated as the proportion of days in the $i$-th month during the window period of $j$-th dressing. Table S1 lists the basal and topdressing fertilization dates for different crops in different provinces.

An example: in 2016, winter wheat was sown and basal fertilizer was applied in late September to mid-October. So, the dates of basal dressing in Hebei province span two months; $x$=1/3 in September, while $x$=2/3 in October. Similarly, the other two top dressings were applied at the jointing and booting stages, i.e., in late March to early April and late April to early May, respectively. According to the proportion of basal dressing and top dressing for wheat in Hebei, we identified the proportion of total fertilizer applied in each month (basal dressing: 0.2, September; 0.4, October; top dressing: 0.1, March; 0.2, April; and 0.1 in May). Further, the proportion of fertilizer applied in each month in Sichuan, which is located in Southwest China, was different from that in Hebei (basal dressing: 0.2, October; 0.4, November; top dressing: 0.1, January; 0.2, February; and 0.1, March). Besides, four sub-categories of rice were considered in this study (early, middle, late, and single-season rice), and given the differences in their fertilization dates in different regions, the fertilization dates varied greatly across the country. Furthermore, a comprehensive assessment of the fertilizer application times and dates corresponding to each crop is of great significance with respect to improving the accuracy of $NH_3$ emission inventories. Therefore, details regarding the fertilization months and the fertilization ratios corresponding to basal dressing and top dressing for the three main crops were considered (Fig. 1). For other crops, we also identified the corresponding fertilization dates and ratios according to their respective phenological periods by collecting large amounts of data from existing literatures and reports (Zhang et al., 2009; Zhang and Zhang, 2012; Zhang et al., 2018).

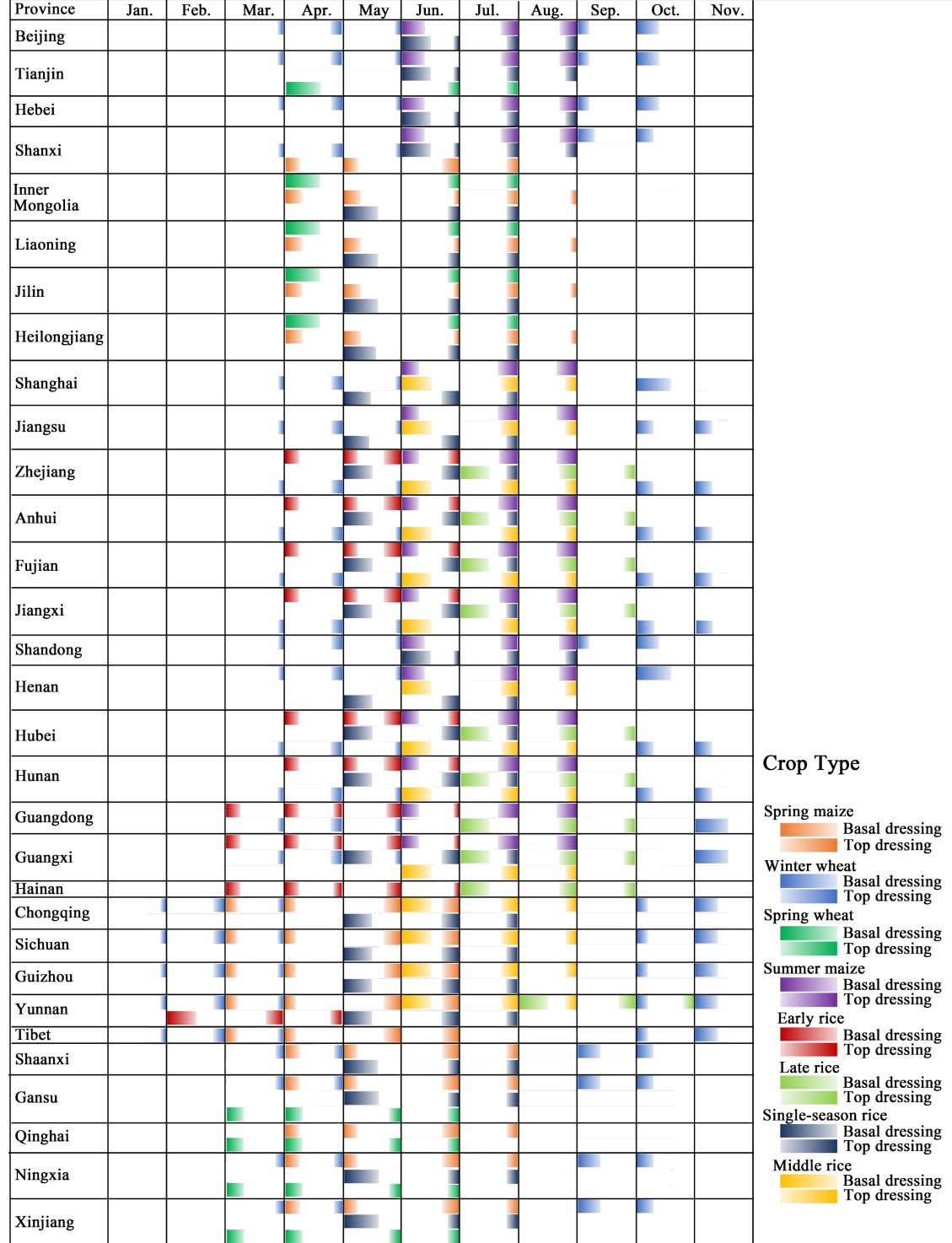

**Fig. 1. Basal and topdressing fertilization months and ratios for maize, wheat, and rice in China.**

(2) Given that rice, which is widely cultivated across China, requires N fertilization, the excessive use of N fertilizer results in sizeable NH$_3$ emissions (Xia et al., 2020). Fig. 1 shows that compared with other crops, the amount of fertilizer applied to rice in different months is more complex. This implies that more attention should be paid to the spatial allocation of fertilizer quantity corresponding to rice. Therefore, using the spatial distribution of each rice sub-category (i.e., early, late, single-season, and middle rice) as spatial proxies, rather than the overall spatial distribution of rice, can greatly minimize the spatial bias of NH$_3$ emissions from rice fertilization applications. In this study, we used a variety of data to integrate the spatial distribution of the four rice types in the country. First, spatial distribution data corresponding to the abovementioned rice types in ten provinces of southern China (Henan, Jiangsu, Anhui, Hubei, Hunan, Jiangxi, Zhejiang, Fujian, Guangdong, and Shanghai) were directly obtained from (Qiu et al., 2015). This dataset was proposed based on 500 m 8 day composite Moderate Resolution Imaging Spectroradiometer (MODIS) Enhance Vegetation Indices with two bands (EVI2). Its efficiency was validated using data from 763 ground survey sites, and it showed an overall accuracy of 95.02%. Second, for other provinces, we used land use and land cover change (LUCC) data as well as cropping frequency data. Areas where paddy fields overlap with a single cropping frequency were considered as single-season rice areas. Such areas were primarily distributed in Northeast China. Further, areas where paddy fields overlap with a double cropping frequency were considered double-cropping or middle-rice areas. According to the yield of the two rice types in each province, the spatial distribution of early/late rice and middle rice areas can be further distinguished. LUCC data and cropping frequency data with a resolution of 1 km were downloaded from the Resource and Environment Science and Data Center (http://www.resdc.cn/data.aspx?DATAID=184 and http://www.resdc.cn/DOI/doi.aspx?DOIid=42). The spatial distributions of the rice types (single-season rice, middle rice, and early/late rice) are shown in Fig. S1. For other crops, we used the EarthStat dataset on crop harvest area (Monfreda et al., 2008), which provides global crop harvest areas and yields at a 5 min × 5 min resolution.

After completing the above improvements, we further considered the effects of soil properties, agricultural activity, and meteorological conditions (Bouwman et al., 2002; Zhang et al., 2018). The monthly emission factors corresponding to fertilizer application were calculated as follows:

$$EF = EF_0 \times e^{f_{pH} + f_{CEC} + f_{crop} + f_{method}} \times \alpha \tag{3}$$

where $EF_0$ represents the baseline emission factors that reported by (Cai et al., 2002; Dong et al., 2009; Zhou et al., 2016) (Table S2), $f$ represents the effects of soil pH, soil cation exchange capacity (CEC), fertilization types (basal dressing and top dressing), and crop types (upland crops and paddy field crops) based on (Bouwman et al., 2002; Zhang et al., 2018). Soil pH and CEC data were obtained from the Harmonized World Soil Database (http://www.fao.org/land-water/databases-and-software/hwsd/en/). Detailed $f$ values are listed in Table S2. Further, α represents the monthly scalar, which was applied to characterize the influence of meteorological factors on NH$_3$ emissions (Gyldenkaerne et al., 2005; Zhang et al., 2018).

$$\alpha = (e^{0.0223T_i + 0.0419W_i}) / (\frac{1}{12} \sum_{i=1}^{12} e^{0.0223T_i + 0.0419W_i})$$

(4)

where $T_i$ and $W_i$ represent 2 m air temperature (°C) and 10 m wind speed (m/s) for a given month, $i$, respectively. T and W were processed using ECMWF ERA5 Reanalysis data (https://www.ecmwf.int/en/forecasts/datasets/reanalysis-datasets/era5).

## 2.2 NH₃ emission from livestock waste

Traditional NH$_3$ emissions from livestock waste are usually calculated as a product of livestock population and the corresponding emission factors. In this study, a more process-based mass-flow approach was applied, considering nitrogen
transformation at the different stages of manure management (Huang et al., 2012; Kang et al., 2016; EEA, 2019). The total ammoniacal nitrogen (TAN) amount was obtained using the annual livestock amount, the daily amount, and the nitrogen content of urine and feces for each livestock category. Details in this regard are provided in Table S3. The outdoor and indoor generated TAN contents were separately estimated based on the proportion of the time each livestock category spent indoor and outdoor, respectively. There are three main livestock breeding systems in China, namely grazing, free-ranging, and
intensive livestock breeding systems. Among them, half of the livestock urine and feces corresponding to the grazing and free-range systems are discharged indoors. However, for intensive livestock breeding, all the livestock urine and feces are discharged indoors (MEP, 2014). Grazing is practiced only in pastoral and semi-pastoral areas, i.e., in 13 provinces (Hebei, Shanxi, Inner Mongolia, Liaoning, Jilin, Heilongjiang, Sichuan, Yunnan, Tibet, Gansu, Qinghai, Ningxia, and Xinjiang). According to the nitrogen flow and phase of manure management, the activity levels were classified under seven categories:
outdoor, housing solid, housing liquid, storage solid, storage liquid, spreading solid, and spreading liquid. Thus, NH$_3$ emissions from livestock were calculated as a product of the TAN of the seven categories and the corresponding emission factors (Huang et al., 2012; MEP, 2014; Kang et al., 2016). Livestock production in each province was obtained from MARA (2017) and NBS (2017c).

After estimating the NH$_3$ emissions corresponding to the three livestock breeding systems in different provinces, we
allocated the grazing emissions to grids based on grassland areas in the pastoral and semi-pastoral areas and allocated the emissions from free-range and intensive livestock production based on the corresponding rural residential areas. In addition to emissions from fertilizer applications, the influence of meteorological factors on NH$_3$ emissions from livestock waste was also considered. For outdoor NH$_3$ emissions, we considered the influence of monthly temperature and wind speed using Eq. (3), while accounting only for air temperature for indoor emissions (Zhang et al., 2018).

## 2.3 NH₃ emission from other sources

NH$_3$ emissions from combusted crop residue were estimated based on crop yield, grain-to-straw ratio, combustion ratio, and combustion efficiency (Zhou et al., 2017). The contribution of firewood combustion to NH$_3$ emissions was derived from (Cong et al., 2017). Additionally, grassfire and forest fire data were obtained by coupling MCD14ML and MCD64A1 fire products

based on previous studies (Qiu et al., 2016; Li et al., 2018). The emissions originating from human excrement were calculated based on the daily excretion data corresponding to children and adults, rural populations, and the fraction of tatty latrines in each province (Huang et al., 2012). The emissions corresponding to the total mileage for each vehicle category were calculated using the number of vehicles and the average annual mileage. Further, $NH_3$ emissions from other sources were determined following the approaches proposed by previous studies (Huang et al., 2012; Kang et al., 2016). Relevant data were obtained from NBS, 2017a, b, c, and d. The specific emission factors and spatial allocation methods for each source are listed in Table S4.

## 2.4 $NH_3$ emission inventory uncertainty and accuracy evaluation

The uncertainty of $NH_3$ emissions was calculated using the Monte Carlo method, which has been widely used in various inventory studies (Zhao et al., 2011; Kang et al., 2016; Li et al., 2019a). Based on previous studies, we assumed that the uncertainties in the activity levels and emission factors were uniformly and normally distributed. The detailed parameters of the CVs of the two datasets were derived from Huang et al. (2012). The $NH_3$ emission calculations were replicated 10,000 times with a random selection of all the inputs.

Additionally, the $NH_3$ emission inventory established in this study and the MEIC inventory were applied to WRF-Chem to evaluate the inventory accuracy. Emissions of other air pollutants in 2016 (including $SO_2$, $NO_x$, CO, $CO_2$, NMVOC, BC, OC, $PM_{2.5}$, and $PM_{10}$) were obtained from MEIC (http://meicmodel.org/), with a horizontal resolution of 0.25° (Li et al., 2017b). Simulations were conducted for January, April, July, and October in 2016, to represent the four typical seasons. The simulation domain with a resolution of 27 km is shown in Fig. 2, which covers most parts of North, East, Central, and South China. Areas in China with high $NH_3$ emission density are shown in Fig. 4. Inventory accuracy was further assessed using WRF-Chem and available ground station- and satellite-based observations. First, we compared the $NH_3$ VCDs obtained using infrared atmospheric sounding interferometer (IASI) satellite observations and with those from WRF-Chem using our estimated inventory and MEIC for the four selected months. Daily IASI $NH_3$ VCDs were downloaded from the ESPRI data center (https://cds-espri.ipsl.upmc.fr/etherTypo/index.php?id=1700&L=1). The mean local solar overpass times were 9:30 am and 9:30 pm at the equator. Furthermore, in this study, only the IASI $NH_3$ VCDs collected from the morning orbit were considered, as they are generally more sensitive to $NH_3$ emissions owing to their higher thermal contrast (Van Damme et al., 2014; Van Damme et al., 2015). Furthermore, to ensure comparison accuracy, the average simulated $NH_3$ concentrations at 09:00 and 10:00 local time were applied to calculate the simulated $NH_3$ VCDs (Zhao et al., 2020). Second, we compared the $NH_3$ concentrations simulated using WRF-Chem to ground observation data, in January, April, and July 2016 (as ground observation data were not available for October 2016). Specifically, observed $NH_3$ concentrations were obtained from the Ammonia Monitoring Network in China (AMoN-China) (Pan et al., 2018), and the simulation domain of WRF-Chem consisted of 30 sampling stations from AMoN-China (Fig. 2).

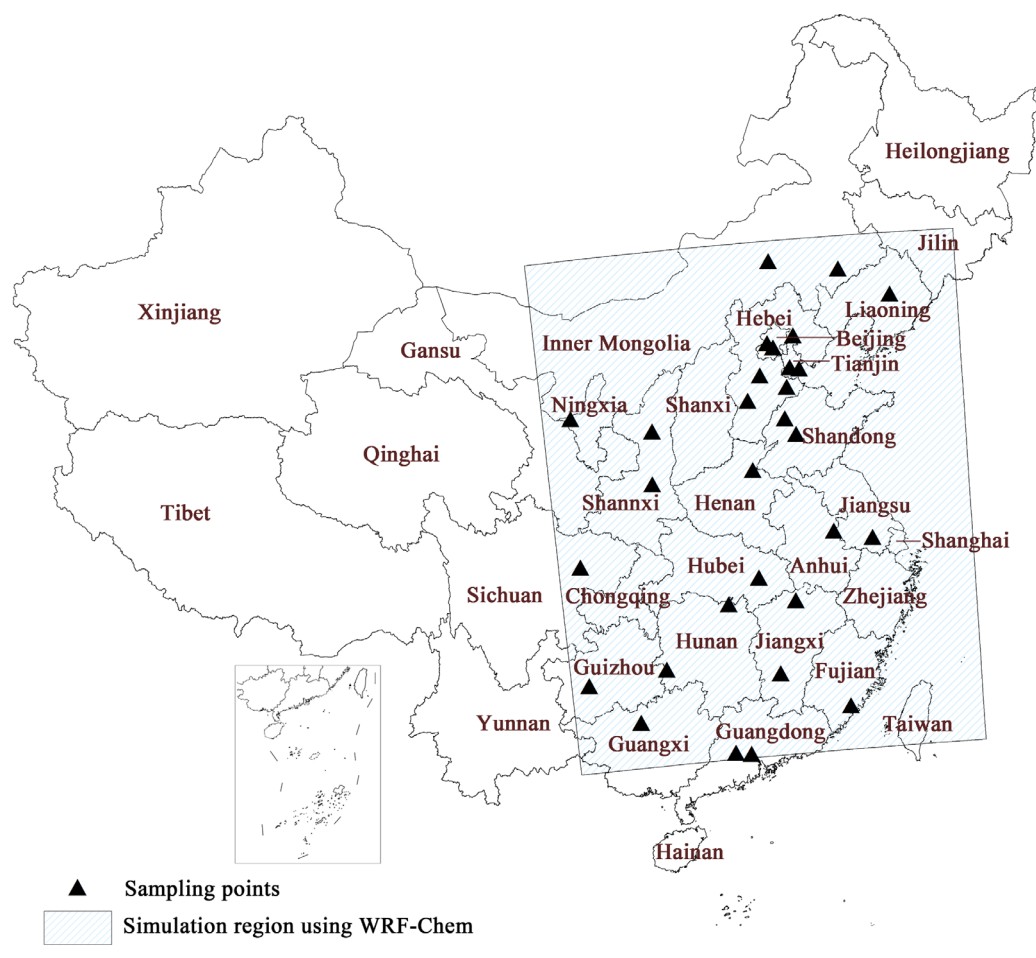

**Fig. 2. WRF-Chem simulation domain and sampling points (Ammonia Monitoring Network, China).**

## 3 Results and discussion

### 3.1 Emissions and main sources of NH₃ in China

In 2016, the total atmospheric ammonia emission in mainland China was 12.11 Tg (10.54−16.04 Tg, 95% confidence interval

based on a Monte Carlo simulation), and the emission density was 1.28 t/km². This total NH₃ emission amount was found to

be approximately three-fold that obtained for Europe (4.18 Tg) (Backes et al., 2016), and contributed approximately 38 and

27% of Asian and global NH₃ emissions, respectively (Bouwman et al., 1997; Kurokawa and Ohara, 2020). Further, this

estimated emission was relatively close to the improved emissions based on AMoN-China and the Ensemble Kalman Filter

(13.1 Tg) (Kong et al., 2019). It was also close to the improved bottom-up emission (11.7 Tg) reported by Zhang et al. (2018).

However, the estimation was approximately 25% higher than that reported by Kang et al. (2016), and approximately 15%

lower than the REAS emission Kurokawa and Ohara (2020). Table 2 presents a quantitative comparison of this emission inventory with those reported in previous studies.

Table 2. Comparison of estimated $NH_3$ emissions with those obtained in other studies.

| Source | Base year | Total | References |
|--------|-----------|-------|-----------|
| EDGARv5.0 | 2015 | 8.9 | https://edgar.jrc.ec.europa.eu/dataset_ap50 |
| REASv3 | 2015 | 14.1 | (Kurokawa and Ohara, 2020) |
| MEIC | 2016 | 10.3 | http://meicmodel.org/ |
| MASAGE_NH3 | 2007 | 8.4 | (Paulot et al., 2014) |
| Zhang et al. | 2015 | 15.6 | (Zhang et al., 2017) |
| Huang et al. | 2006 | 9.8 | (Huang et al., 2012) |
| Xu et al. | 2008 | 8.4 | (Xu et al., 2016) |
| Xu et al. | 2010 | 10.7 | (Xu et al., 2015) |
| Kang et al. | 2012 | 9.7 | (Kang et al., 2016) |
| Kong et al. | 2016 | 13.1 | (Kong et al., 2019) |
| Zhang et al. | 2008 | 11.7 | (Zhang et al., 2018) |
| Streets et al. | 2000 | 13.6 | (Streets et al., 2003) |
| Zhao et al. | 2010 | 9.8 | (Zhao et al., 2013) |
| This study | 2016 | 12.1 | This study |

Similar to other studies, livestock waste (5.42 Tg) and fertilizer application (4.67 Tg) were identified as the two largest $NH_3$ emission sources in China, with their contribution to the total $NH_3$ emission amount reaching over 80% (Fig. 3). Other sources included synthetic ammonia (0.43 Tg), agriculture soil (0.29 Tg), indoor biomass combustion (0.30 Tg), domestic coal combustion (0.26 Tg), nitrogen fertilizer production (0.22 Tg), transportation (0.10 Tg), and others (0.4 Tg), with contributions
to the total emission amount as follows: 3.5, 2.4, 2.5, 2.2, 1.8, 0.9, and 3.3%, respectively.

Similar to previously reported results, with respect to the different fertilizers considered, urea was identified as the major contributor to $NH_3$ emissions, accounting for approximately 45.4% of fertilizer type-related $NH_3$ emissions (Fig. 3). Kang et al. (2016) observed that ABC is an important source of fertilizer type-related $NH_3$ emission source owing to its high volatility; however, in 2016, it only accounted for 12.9% of fertilizer type-related $NH_3$ emissions. This is because, in recent
250 years, there has been a significant decrease in the proportion of ABC-related emissions in China owing to the decrease in the amounts of ABC applied to the three main crops, maize, rice, and wheat (from 26.55 kg/$hm^2$ in 2011 to 10.80 kg/$hm^2$ in 2016, i.e., a 59.32% reduction). However, emissions related to complex fertilizers have increased by 33.61% (NDRC, 2017). Specifically, NPK fertilizers have become the second-largest source of fertilizer-related $NH_3$ emissions in China (26.7%). Therefore, replacing ABC with complex fertilizers might reduce fertilizer-related $NH_3$ emissions.

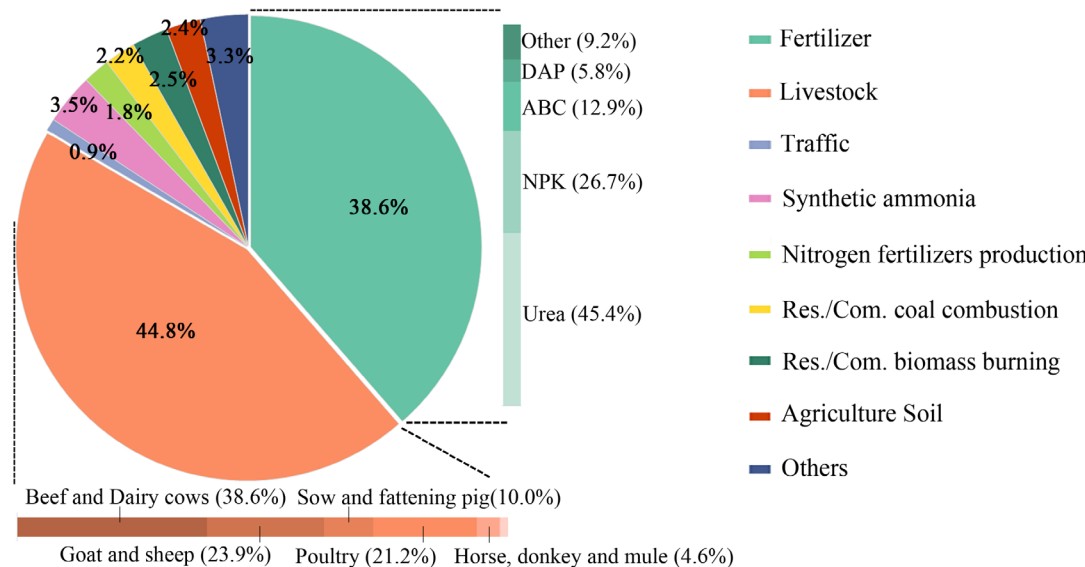

**Fig. 3. Contributions of different sources (%) to NH₃ emissions in mainland China (2016).**

Maize fertilization contributed the most to NH$_3$ emissions (1.21 Tg), accounting for 25.94% of fertilizer application-related emissions. This observation could be primarily attributed to the planting area of maize—the largest of the total crop area, up to 22.06% (NBS, 2017c). Moreover, the maize fertilization dates were concentrated in summer, and the high temperature further increased the NH$_3$ emission rate. It is also worth noting that vegetable fertilization has become the second-largest fertilizer application-related NH$_3$ emission source (0.89 Tg), i.e., 19.06%, based on our estimation. In addition to the relatively large area corresponding to vegetable cultivation in China, vegetable-related fertilizer application rates were higher than those corresponding to the three major crops considered in this study (Wang et al., 2018). The amounts of urea, DAP, and NPK used in the fertilization of vegetable-cultivated land were 1.29, 1.77, and 1.40 times those applied in cropland for cultivating the three main crops, respectively. This is due to the lack of scientific fertilization methods for vegetables in China (NDRC, 2017). Further, rice and wheat fertilization accounted for 17.47 and 14.80% of fertilizer application-related emissions, respectively. Furthermore, NH$_3$ emissions corresponding to the above four crop types accounted for more than 75% of the total NH$_3$ emissions from fertilizer application.

Regarding livestock waste, beef and dairy cow waste were the largest contributors (38.6%) to livestock waste-related NH$_3$ emissions. This was followed by goats and sheep waste (23.9%) and poultry waste (21.2%); this is consistent with Kang et al. (2016) . Further, we analyzed animal breeding system-related nitrogen transformation and migration and observed that the contributions of the four manure management stages to NH$_3$ emissions were 1.06, 1.14, 0.85, and 2.73 Tg for outdoor (19.60%), housing (21.10%), manure storage (15.65%) and manure spreading (43.65%), respectively. This is also consistent with the results of Xu et al. (2017). Our analysis also showed that the free-range system is the largest contributor to livestock

waste-related $NH_3$ emissions, i.e., 2.80 Tg, which accounted for 51.6% of the total livestock waste-related emissions. This was followed by the intensive system (1.92 Tg, 35.47%), and lastly, the grazing system (0.70 Tg, 12.90%). The rapid increase in the proportion of intensive livestock raising has slowed down $NH_3$ emissions to a certain extent (Qian et al., 2018).

## 3.2 Geographical distribution of $NH_3$ emissions

The spatial distribution of $NH_3$ emissions in 2016 is shown in Fig. 4, from which a strong spatial variability is evident. The highest $NH_3$ emission density of 6.96 t/km$^2$, which was 5.44-fold higher than the national average (1.28 t/km$^2$), was observed in Shandong. Furthermore, 21 provinces had $NH_3$ emissions above this national average density, and the provinces with emission densities exceeding 3 t/km$^2$ included: Shandong (6.96 t/km$^2$), Henan (6.81 t/km$^2$), Jiangsu (5.01 t/km$^2$), Tianjin (4.43 t/km$^2$), Hebei (4.39 t/km$^2$), and Anhui (3.41 t/km$^2$). Although these six provinces account for only 8.08% of the total land area

of mainland China, they contributed 33.74% of $NH_3$ emissions in China. These provinces are all concentrated in the North China Plain (NCP, which includes the above six provinces and Beijing), an area that features well-developed crop farming and animal husbandry. The NCP contains a large amount of high-quality arable land, and the farms in this area produce 34.47% of China's major farm products (NBS, 2017c). The fertilization of crops emits large amounts of $NH_3$. The soil in this area is alkalescent (the average pH value is 7.15), which further increases the $NH_3$ volatilization. Additionally, the NCP produces as

much as 28.16 million tons of pork and beef per year, accounting for 32.5% of the total (NBS, 2017c), contributing to higher $NH_3$ emissions in the North China Plain.

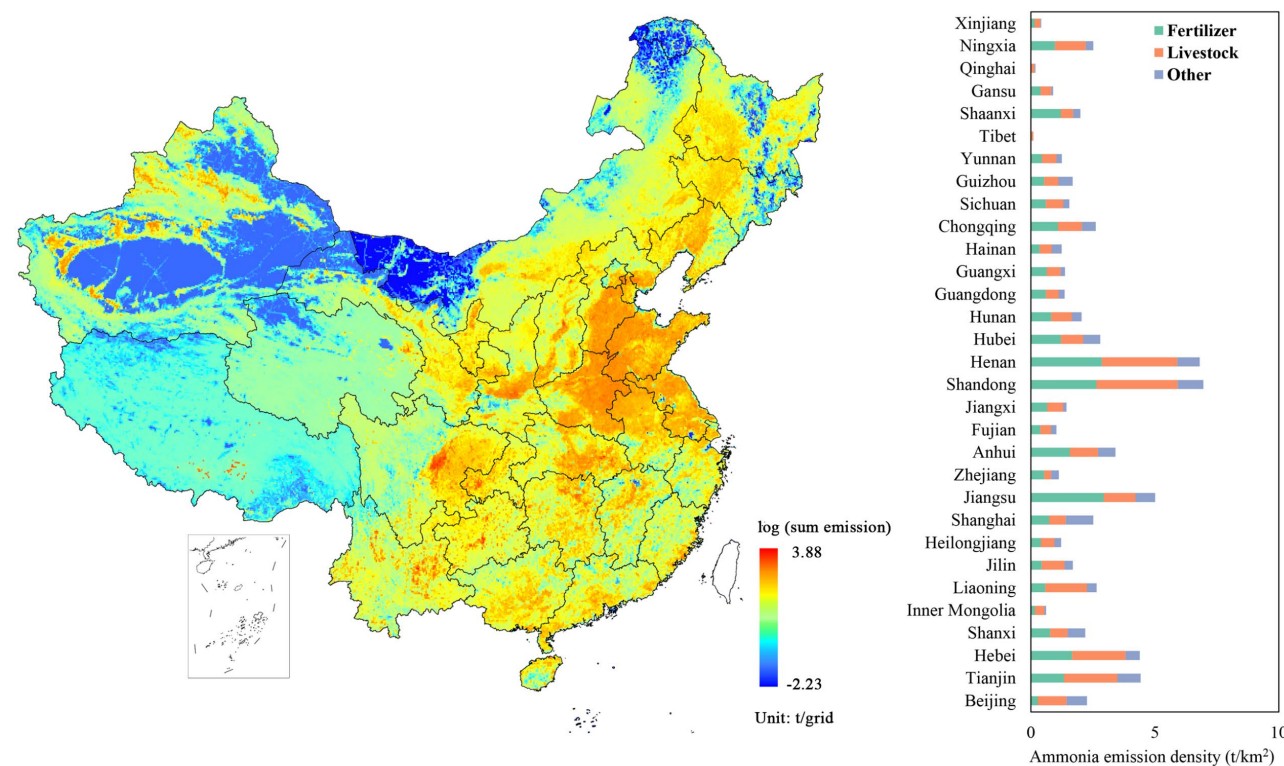

**Fig. 4. Geographical distribution of 2016 NH₃ emissions and emission densities in China.**

Shandong and Henan provinces not only had the two top emission densities, but their emissions were also the top two in mainland China, reaching 1.13 and 1.08 Tg, respectively. These high emissions could be primarily attributed to the following reasons: First Shandong and Henan are the major agricultural provinces in China. Although they represent only 3.40% of the total land area of mainland China, in 2016, they accounted for 38.43, 17.74, and 16.31% of wheat, maize, and vegetables yields, respectively (NBS, 2017c). The NH₃ emissions corresponding to these three crops were remarkable, accounting for 80.35 and 79.23% of the fertilizer application-related NH₃ emissions in Shandong and Henan, respectively. Second, there are a large number of livestock farms in these two provinces (Hu et al., 2017). Specifically, beef cattle and poultry breeding in the two provinces also resulted in high NH₃ emissions from livestock waste, and only these two provinces showed livestock waste-related emission densities above 3 t/km².

Furthermore, Jiangsu province had the highest fertilizer application-related emission density (2.94 t/km²) in China, which was 5.96 times above the national average. This is not only due to the relatively large crop area in this province, but also, more importantly, the fertilizer application rates in this province were much higher than in other provinces. For example, the rates of urea application to wheat and rice, which together constitute the largest planting area in Jiangsu Province, were 191.40 and 223.80 kg/hm², respectively, i.e., 1.42- and 1.97-fold higher than the national average value, respectively. Further,

emissions from fertilizer application accounted for 58.68% of the total $NH_3$ emissions in Jiangsu. Therefore, enhancing nitrogen fertilizer use efficiency is an important and effective strategy by which $NH_3$ emissions in Jiangsu can be controlled.

The $NH_3$ emissions in Sichuan and Xinjiang in Western China were also high, ranking fourth (0.75 Tg) and sixth (0.68 Tg) in the country, respectively. Specifically, in Sichuan, rice and vegetable cultivation was identified as the principal fertilizer application-related $NH_3$ emission sources, (25.73 and 20.85%, respectively), while cotton cultivation was identified as the main emission source in Xinjiang (49.20%). These results reflect the huge differences in planting structure in China. In terms of livestock breeding, the largest emission sources in Sichuan were beef cattle and dairy cow breeding (25.43%) and goat and sheep breeding (23.79%), while the total contribution of these two livestock breeding practices in Xinjiang was 86.57%, indicating significant differences in dietary habits in the different provinces China. These findings also indicate that different livestock waste-related $NH_3$ emission reduction strategies are needed for different regions. The specific $NH_3$ emissions corresponding to different emission sources in the different provinces are shown in Table 3, and the spatial distribution of $NH_3$ emissions from fertilizer application, livestock waste, and other sources are shown in Fig. 4.

**Table 3. NH₃ emissions (Gg) from various sources in the different provinces in mainland China.**

| Province | Fertilizer | | | | | Livestock | | | Other |
|---|---|---|---|---|---|---|---|---|---|
| | Urea | ABC | DAP | NPK | Other | Free | Grazing | Intensive | |
| Beijing | 2.08 | 0.05 | 0.36 | 1.79 | 0.24 | 4.10 | 0.00 | 15.02 | 13.53 |
| Tianjin | 8.48 | 0.62 | 1.94 | 4.10 | 0.78 | 7.75 | 0.00 | 17.98 | 11.14 |
| Hebei | 144.97 | 17.68 | 28.20 | 107.11 | 15.08 | 185.17 | 20.51 | 199.82 | 107.78 |
| Shanxi | 58.93 | 8.96 | 10.48 | 15.15 | 28.56 | 48.43 | 1.41 | 60.05 | 112.35 |
| Inner Mongolia | 108.06 | 34.72 | 30.99 | 9.97 | 8.74 | 40.72 | 242.87 | 148.32 | 79.26 |
| Liaoning | 34.11 | 1.64 | 6.89 | 28.43 | 13.93 | 118.52 | 21.97 | 106.43 | 56.99 |
| Jilin | 37.45 | 0.85 | 7.76 | 17.86 | 19.93 | 86.25 | 25.61 | 64.39 | 63.50 |
| Heilongjiang | 102.53 | 0.00 | 25.23 | 37.76 | 18.22 | 94.23 | 46.40 | 105.97 | 123.07 |
| Shanghai | 3.76 | 0.28 | 0.02 | 1.75 | 0.17 | 1.92 | 0.00 | 3.47 | 8.90 |
| Jiangsu | 178.11 | 15.06 | 2.83 | 86.35 | 19.56 | 50.58 | 0.00 | 80.92 | 81.08 |
| Zhejiang | 27.87 | 4.50 | 0.04 | 17.26 | 6.86 | 16.90 | 0.00 | 12.97 | 32.35 |
| Anhui | 107.86 | 5.00 | 1.04 | 82.95 | 24.12 | 106.16 | 0.00 | 52.50 | 98.85 |
| Fujian | 9.34 | 17.95 | 0.01 | 12.18 | 7.00 | 24.37 | 0.00 | 28.25 | 27.29 |
| Jiangxi | 44.85 | 4.27 | 0.04 | 46.38 | 14.21 | 80.49 | 0.00 | 27.27 | 23.39 |
| Shandong | 130.75 | 24.10 | 20.42 | 196.05 | 40.66 | 258.12 | 0.00 | 252.73 | 162.09 |
| Henan | 165.76 | 59.08 | 9.18 | 174.22 | 65.62 | 340.50 | 0.00 | 164.65 | 148.63 |
| Hubei | 52.13 | 78.79 | 0.63 | 65.15 | 29.63 | 92.95 | 0.00 | 72.38 | 128.85 |
| Hunan | 61.72 | 36.60 | 0.31 | 44.12 | 31.02 | 137.98 | 0.00 | 40.30 | 81.46 |
| Guangdong | 46.33 | 9.94 | 0.08 | 50.38 | 2.04 | 66.26 | 0.00 | 24.40 | 43.17 |
| Guangxi | 65.03 | 9.89 | 0.40 | 59.97 | 17.49 | 111.96 | 0.00 | 18.50 | 42.76 |
| Hainan | 4.50 | 0.09 | 0.00 | 6.39 | 1.29 | 14.91 | 0.00 | 2.08 | 13.99 |
| Chongqing | 27.76 | 24.63 | 0.05 | 31.70 | 6.19 | 68.94 | 0.00 | 10.62 | 45.50 |
| Sichuan | 103.66 | 106.80 | 0.18 | 60.43 | 23.38 | 231.96 | 44.73 | 62.37 | 119.67 |
| Guizhou | 57.67 | 4.47 | 0.31 | 18.49 | 13.43 | 82.93 | 0.00 | 17.67 | 102.41 |
| Yunnan | 109.41 | 23.92 | 1.29 | 24.32 | 12.81 | 189.80 | 1.63 | 35.94 | 78.72 |
| Tibet | 0.50 | 0.38 | 0.01 | 0.15 | 0.09 | 54.33 | 52.32 | 5.84 | 6.94 |
| Shaanxi | 110.84 | 85.87 | 21.93 | 27.75 | 1.55 | 77.98 | 0.00 | 24.59 | 59.27 |
| Gansu | 112.96 | 5.27 | 27.01 | 13.36 | 6.88 | 70.96 | 50.25 | 62.81 | 32.24 |
| Qinghai | 5.79 | 0.52 | 2.21 | 0.41 | 0.17 | 3.76 | 69.85 | 19.27 | 25.99 |
| Ningxia | 18.76 | 19.07 | 7.85 | 4.93 | 0.00 | 15.96 | 18.16 | 30.46 | 15.56 |
| Xinjiang | 177.39 | 1.10 | 62.57 | 2.61 | 1.48 | 114.87 | 103.78 | 155.04 | 64.84 |
| Sum | 2119.36 | 602.09 | 270.23 | 1249.44 | 431.14 | 2799.77 | 699.49 | 1923.02 | 2011.59 |

## 3.3 Monthly variation of NH₃ emissions

The monthly variation of NH₃ emissions from the main emission sources is shown in Fig. 5. Unlike some pollutants (such as

PM$_{2.5}$ and BC) which exhibit higher emissions in winter, NH₃ emissions tended to be high in summer and low in winter. The

highest and lowest emissions (1.68 and 0.55 Tg, respectively) were recorded in July and January, respectively; the July to January emission ratio was 3.08, which is close to the ratio obtained based on IASI satellite observations (2.85), and larger than that based on MEIC data (1.72).

Further, a comparison of the monthly trend of $NH_3$ emissions based on IASI satellite observations with that obtained in this study showed a strong correlation between the two trends ($R^2 = 0.85$). Owing to temperature changes, livestock waste-related emissions increased slowly from 0.30 Tg in January to 0.60 Tg in July, and then gradually decreased to 0.34 Tg in December. Additionally, the monthly fluctuation of fertilizer application-related $NH_3$ emissions was greater than that of livestock waste-related emissions (Fig. 5(a)). Fertilizer application-related $NH_3$ emissions in July were 11.79 times than those observed in January. These monthly variations in $NH_3$ emission amounts could be primarily attributed to the effect of the farming season on fertilization. Thus, we further analyzed monthly $NH_3$ emission variation with respect to different crops (Fig. 5(b)). The results obtained was observed that maize and rice fertilization in summer was primarily responsible for the increase in $NH_3$ emissions in China. For example, $NH_3$ emissions from maize and rice fertilization in July accounted for 44.89 and 27.61% of the total fertilizer application-related $NH_3$ emissions, followed by cotton fertilization (8.59%). Even though wheat also contributed significantly to $NH_3$ emissions (14.80%), the emissions were primarily concentrated in April and September, accounting for 36.38 and 23.47% of the emissions from fertilizer application in these months, respectively.

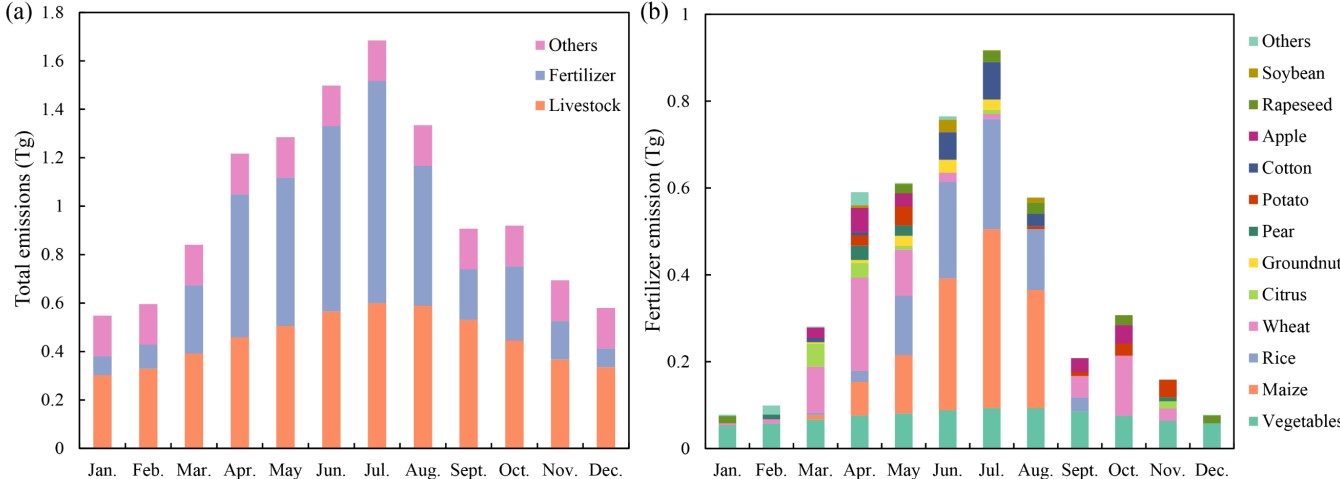

**Fig. 5. Monthly NH₃ emissions from: (a) Different sources and (b) Different crops.**

## 3.4 Uncertainty and accuracy of the NH₃ emission inventory

### 3.4.1 Uncertainty

Uncertainty in the estimated $NH_3$ emissions results from both the activity level and EF input data. We ran 10000 Monte Carlo simulations to estimate the range of $NH_3$ emissions from each source with a 95% confidence interval. The estimated total $NH_3$ emission range was 10.5-16.0 Tg. The 95% confidence intervals of fertilizer application, livestock waste, and others ranged

from -20.5% to 64.41%, -23.0% to 37.1% and -42.9% to 62.4% (Fig. 6). Due to the large amounts of NH$_3$ emitted by fertilizers and livestock waste, the uncertainty of total NH$_3$ emissions is mainly caused by the uncertainties of these two sources. The
uncertainty of fertilizer application was slightly greater than that of livestock waste. The emission factors, especially the corrected EF, were the largest contributors to the uncertainties of fertilizer application emissions. Additionally, it is clear that NH$_3$ emissions from other sources exhibited the largest uncertainty (-42.9% to 62.4%), mainly due to the high degree of uncertainty resulting from the many sub-sources, such as $-77.1\% \sim 96.9\%$ of the transportation sector and $-79.4\% \sim 122.7\%$ of the industrial sector. In comparison, the emissions from other sources were relatively small; hence, the large uncertainties
of other sources did not have a significant impact on the uncertainty of total NH$_3$ emissions.

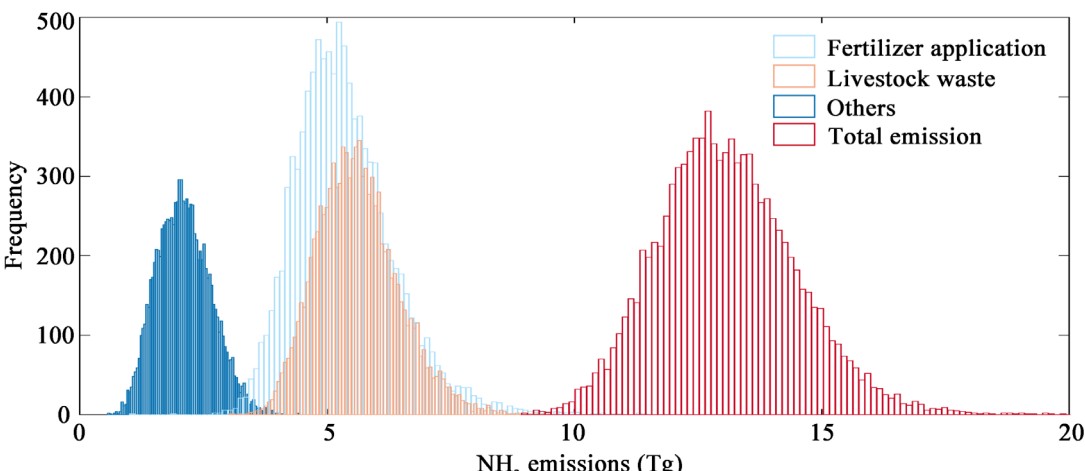

**Fig. 6. Uncertainties of NH$_3$ emissions sourced from fertilizer application, livestock waste, and others.**

### 3.4.2 Temporal accuracy of NH$_3$ emission inventory

We verified the accuracy of the monthly NH$_3$ emission trends obtained in this study by determining the correlation between
the monthly IASI satellite observations and the monthly emissions of two inventories (MEIC and our study) in 2016. It was observed that the monthly trend of our inventory was significantly more correlated with the IASI data (R$^2$ = 0.84) than the MEIC inventory (R$^2$ = 0.70). The correlations of the monthly trends of the IASI data and the two inventories were also compared for different regions in China. The results showed that except for Central China, the correlations of the monthly trends between our inventory and IASI data were higher than those between the MEIC inventory and IASI data (Table 4).
Additionally, the July to January emission ratio was used to further verify inventory accuracy, and it was observed that our emission inventory ratios were close to those based on IASI data for all regions of China. However, the ratios based on MEIC data were relatively lower, possibly owing to the underestimation of NH$_3$ emissions in summer.

**Table 4. Comparison of the monthly NH₃ emission trends corresponding to IASI satellite observations with the two inventories.**

| Region | $R^2$ | | July/Jan ratio | | |
|---|---|---|---|---|---|
| | IASI vs. OUR | IASI vs. MEIC | IASI | OUR | MEIC |
| Northeast China | **0.92** | 0.67 | 3.81 | 3.94 | 1.66 |
| Northern China | **0.79** | 0.69 | 2.77 | 3.51 | 1.66 |
| Eastern China | **0.62** | 0.53 | 4.63 | 3.51 | 1.81 |
| Southern China | **0.58** | 0.33 | 3.93 | 2.01 | 1.84 |
| Central China | 0.47 | 0.57 | 4.09 | 2.83 | 1.75 |
| Southwest China | **0.43** | 0.31 | 2.22 | 2.02 | 1.61 |
| Northwest China | **0.79** | 0.59 | 2.16 | 3.58 | 1.77 |
| Beijing-Tianjin-Hebei region | **0.59** | 0.58 | 3.95 | 3.35 | 1.69 |
| Yangtze River Delta region | **0.62** | 0.49 | 5.45 | 3.98 | 1.95 |
| Mainland China | **0.84** | 0.70 | 2.85 | 3.08 | 1.72 |

Note: $R^2$ value was obtained by fitting the 2016 monthly values between IASI satellite observations and the NH₃ emissions from two
inventories (MEIC and our study). The $R^2$ values in bold represent improved monthly NH₃ emission trend in our inventory compared to that
of the MEIC inventory in the corresponding regions.

### 3.4.3 Spatial accuracy of our NH₃ emission inventory

We compared the simulated NH₃ concentration using WRF-Chem to ground observations obtained from AMoN-China in
January, April, and July (Fig. 7(a)). Thus, it was observed that the spatial accuracy of our inventory was better than that of the
MEIC inventory. Additionally, the simulated NH₃ concentration based on our inventory showed high correlation with ground-
based observations ($R^2 = 0.52$), with a slope of 0.9 (close to 1). However, the correlation between the simulated concentration
based on MEIC data and ground-based observations was relatively low, $R^2 = 0.20$, with a slope of only 0.33 (Fig. 7(b)). The
y-intercepts of the trend lines between the ground observations and the simulated measurements using MEIC and our inventory
were respectively 5.0 and 4.0. Positive intercepts mainly resulted from simulation results that overestimated the NH₃
concentrations at relatively low values (Fig. 7(b)) (e.g., the simulation for January). However, the MEIC inventory may have
underestimated the NH₃ concentration in areas with a high emission density (e.g., the simulation for the North China Plain in
July). For each month, we found similar results for the spatial accuracy of our inventory, which were better than that of the
MEIC. The $R^2$ values and slopes obtained by fitting the ground-based observations with the NH₃ concentrations simulated via
the MEIC yielded an $R^2$ of 0.18 and slope of 0.54 in January, $R^2$ of 0.01 and slope of 0.11 in April, and $R^2$ of 0.21 and slope
of 0.28 in July, which were significantly lower than those obtained by fitting the simulated NH₃ concentrations using our
inventory and the ground-based observations ($R^2$ of 0.27 and slope of 0.68 in January, $R^2$ of 0.23 and slope of 0.70 in April,
and $R^2$ of 0.53 and slope of 0.87 in July).

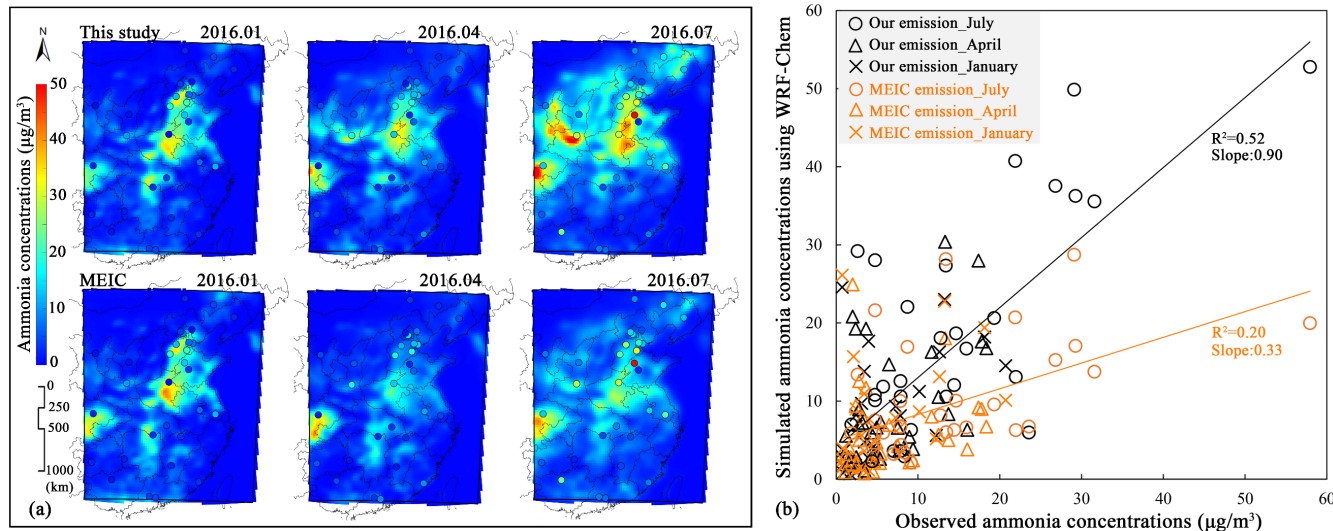

**Fig. 7. (a) Spatial distribution of NH₃ concentrations in 2016, from this study and the MEIC inventory. (b) Correlation**
**between simulated NH₃ concentrations, from different emission inventories, and ground observations obtained from**
**AMoN-China.**

We also compared the NH$_3$ VCDs based on IASI satellite observations with those from WRF-Chem based on the inventory established in this study and that based on MEIC data at a 0.25° resolution. The NH$_3$ VCDs simulated using our inventory exhibited a similar distribution to that of the IASI VCDs (Fig. 8), and through a density scatter plot, were shown to have a better spatial accuracy than that of the MEIC inventory (Fig. S3). In April and July, fitting the NH$_3$ VCDs, simulated with our inventory, with the IASI-based VCDs returned coefficients of determination of 0.41 and 0.42, higher than those obtained by fitting the NH$_3$ VCDs based on MEIC data and IASI data (0.37 and 0.28) (Fig. S3). We observed that the MEIC-based NH$_3$ emissions were significantly underestimated for April and July (a slope of only 0.18 and 0.22). The normalized mean biases (NMBs) between IASI VCDs and MEIC-simulated NH$_3$ VCDs were -76.3% and -42.5% in April and July, respectively. For our simulated NH$_3$ VCDs, they were -64.9 and 6.1%, respectively. Furthermore, in January and October the spatial accuracy of our inventory (NMB: -23.8% in January, -10.4% in October) was better than that of the corresponding MEIC inventory (NMB: -26.5% in January, -22.9% in October).

In general, our inventory exhibited better spatial accuracy than other inventories that utilize WRF-Chem and observations, although January and October showed a relatively large bias between measured and simulated data. The NH$_3$ concentrations were low in these two months. We found that the accuracy of the simulation increased as the concentration of atmospheric NH$_3$ increased. The mean values of IASI NH$_3$ VCDs in the simulation domain in January and October were $7.2 \times 10^{15}$ and $8.7 \times 10^{15}$ molec/cm$^2$, respectively, significantly lower than the mean values of $14.9 \times 10^{15}$ and $19.4 \times 10^{15}$ molec/cm$^2$ in April and July. In January and October, the relatively low NH$_3$ emissions (32.5% and 54.6% of July emissions) combined with the short lifetime of NH$_3$ and uncertainties in gaseous NH$_3$ and aerosol NH$_4^+$ partitioning pose a challenge to the chemical

mechanisms of the WRF-Chem model, making it more difficult for the model to fully capture the heterogeneity of $NH_3$ concentration. Besides, the relatively large uncertainty of the IASI VCDs could also contribute to inconsistency between simulated and observed concentrations (Van Damme et al., 2017; Chen et al., 2020). Zhang et al. (2018) and Zhao et al. (2020) obtained similar results which indicated that the correlations between measured and simulated $NH_3$ data were lower in January and October than during the other two months.

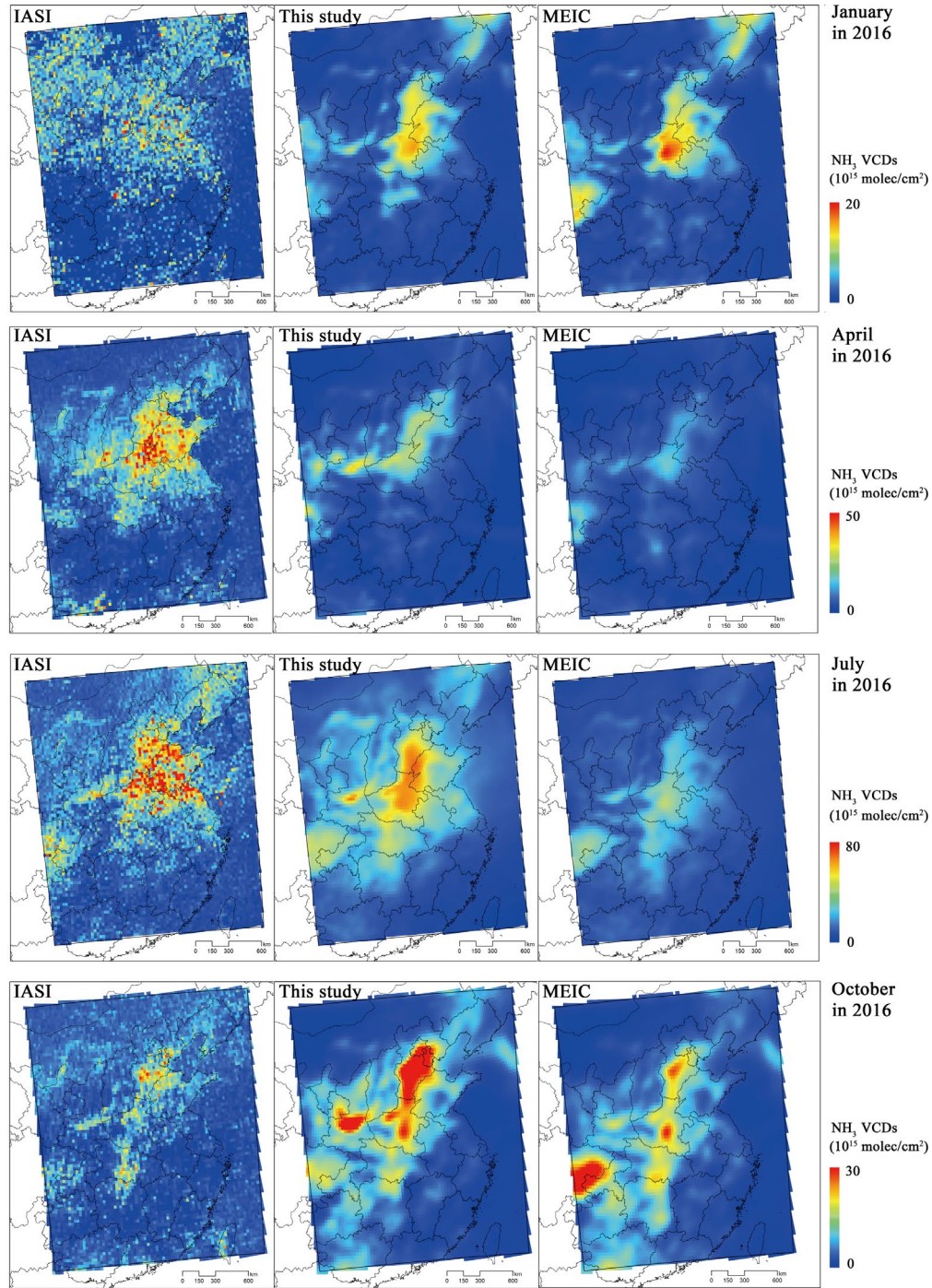

**Fig. 8. The spatial distribution of IASI NH₃ VCDs and NH₃ VCDs from WRF-Chem based on the two inventories in January, April, July, and October (2016). Different color scales represent different months to indicate the spatial distribution of the NH₃ VCDs.**

## 4 Conclusion

NH$_3$—an important component of the nitrogen cycle—can accelerate the formation of SIAs. Even though several effective measures have been taken to reduce SO$_2$ and NO$_X$ emissions, the concentration of NH$_3$ in the atmosphere continues to rise, and unfortunately, gridded NH$_3$ emissions inventories for China still have large uncertainties. Therefore, establishing and improving such gridded NH$_3$ emission inventories can optimize the simulation results of chemical transport models such as WRF-Chem; this is of great significance in regional pollution control. Therefore, in this study, we focused on improving NH$_3$ emission inventories owing to fertilizer application. To this end, we comprehensively evaluated the times and dates of fertilizer application to the major crops that are cultivated in different regions in China, improved the spatial allocation methods for NH$_3$ emissions from different rice types, and established a gridded NH$_3$ emission inventory for mainland China (2016) with a 5 min × 5 min resolution.

The atmospheric ammonia emission in mainland China was found to be 12.11 Tg, and the average emission density was 1.28 t/km$^2$. Livestock waste (44.8%) and fertilization application (38.6%) were identified as the two major NH$_3$ emission sources in China. On the one hand, beef and dairy cow breeding contributed the most to livestock waste-related NH$_3$ emissions, with the free-range system accounting for more than half of the emissions from livestock waste. On the other hand, urea (45.4%) and NPK (26.7%) applications were identified as the main fertilizer application-related NH$_3$ emission sources. NH$_3$ emissions from the cultivation of maize, vegetables, rice, and wheat (25.94, 19.06, 17.47, and 14.8%, respectively) accounted for over 75% of the total emissions from fertilizer application, and in addition to showing the top two emission densities, Shandong and Henan Provinces also showed the top two emissions amounts in mainland China, reaching 1.13 and 1.08 Tg, respectively. The highest emission (1.68 Tg) was recorded in July and the lowest (0.55 Tg) in January. We also observed a strong correlation between the monthly trend of NH$_3$ emissions based on IASI satellite observations and that established in this study ($R^2 = 0.84$). This monthly variation in NH$_3$ emissions was primarily due to the effect of the farming season on fertilization process. Specifically, the fertilization of maize and rice in summer was primarily responsible for the increase in NH$_3$ emissions in China. Additionally, the evaluation of the spatial and temporal accuracies of the NH$_3$ emission inventories obtained in this study using WRF-Chem and AMoN-China observations as well as IASI VCDs indicated that the accuracy of our inventory is better than that of other inventories.

We believe that the improved NH$_3$ emission inventory can be used in future research, to simulate atmospheric aerosol formation, investigate the influence of NH$_3$ emission on PM$_{2.5}$ mass burden and aerosol pH changes, develop targeted NH$_3$ reduction strategies to further improve air quality, and explore the atmospheric N cycle process.

*Data availability.* The gridded ammonia emission inventory is archived on Zenodo (https://doi.org/10.5281/zenodo.5516929; Li, 2021). The ammonia emissions for each source are available upon request from the first author Baojie Li (baojieli@nuist.edu.cn) or the corresponding author Hong Liao (hongliao@nuist.edu.cn).

*Author contributions.* Baojie Li and Hong Liao designed and performed this study. Lei Chen performed the WRF-Chem simulations and data analysis. Jianbing Jin contributed the monthly IASI data. Weishou Shen, Teng Wang, Pinya Wang and Yang Yang discussed the results and commented on the paper.

*Competing interests.* The authors declare that they have no conflict of interest.

*Acknowledgements.* We acknowledge the Centre National d'Études spatiales (CNES, France), and MEIC team for making their data publicly available. We also thank Yuepeng Pan from Institute of Atmospheric Physics, Chinese Academy of Sciences, for publishing the observed $NH_3$ concentrations data (AMoN-China) (Pan et al., 2018), which is helpful for the verification of the $NH_3$ emission inventory.

*Financial support.* This research has been supported by the National Natural Science Foundation of China [grant 42007381], the Natural Science Foundation of Jiangsu Province [grant BK20200812], and the National Key Research and Development Program of China [grant 2020YFA0607803].

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
