# Peer review of "Improved gridded ammonia emission inventory in China"

_Atmospheric Chemistry and Physics, 2021_

## Author Comment (AC1)

General comments:

This study presents an optimized ammonia emission inventory with a focus on fertilizer application relevant sources over mainland China in 2016. The approach adopted in this study considers the time variability of fertilizer application and thus enables this inventory accurate capture the ammonia emissions over time. To illustrate the capability of this emission inventory against existing emission inventory (MEIC developed by Tsinghua University), the authors evaluate the emission inventory by comparing modelled NH3 concentrations by WRF-Chem with ammonia inventory in this work and MEIC. Further, the model performance is validated with NH3 observations from AMoN-China and IASI NH3 column. This paper is well-written and presents results that would be interesting to the air quality modeling community. However, I have several concerns that the authors should consider when revising the manuscript, as listed below. I recommend this work to be published after the following comments are adequately addressed.

Response: We sincerely thank the reviewer for providing valuable and constructive comments. The detailed responses are listed as follows.

Major comments:

1) While the authors illustrated that large uncertainties remain in the NH3 emissions from agriculture sector thus motivates the improvement in NH3 emission inventory, I think vehicular NH3 emissions co-emitted with nitrogen oxides (NOx) are also important sources for NH3. This additional NH3 source has been always underestimated in emission inventory (such as MEIC). Although this work is focused on agriculture ammonia emissions, I would suggest the authors briefly discuss this in the Introduction because this work is entitled "ammonia emission inventory in China".

Response: Thanks for your suggestion. we added some sentences to state the importance of $NH_3$ emissions from the transportation sectors in the Introduction.

"Of course, certain non-agricultural sources of $NH_3$ are also important. For example, vehicular emissions contain both $NH_3$ and $NO_x$ and may have a more effective pathway to particle formation, particularly in urban areas (Farren et al., 2020; Chang et al., 2016). The amount of $NH_3$ emitted by the transportation sector has not been well quantified, and is generally thought to be underestimated (Meng et al., 2017; Farren et al., 2020). "

2) In Sec. 2.4, the uncertainty assessment of NH3 emission inventory established in this work based on the Monte Carlo method is missing. Please provide the uncertainties for each sub-sector if possible.

Response: We are grateful for this critical comment. We added one sub-section (Section 3.4.1) to discuss the uncertainty of $NH_3$ emission based on the Monte Carlo simulation.

**"3.4.1 Uncertainty**
Uncertainty in the estimated $NH_3$ emissions results from both the activity level and EF input data. We ran 10000 Monte Carlo simulations to estimate the range of $NH_3$ emissions from each source with a 95% confidence interval. The estimated total $NH_3$ emission range was 10.5-16.0 Tg. The 95% confidence intervals of fertilizer application, livestock waste, and others ranged from -20.5% to

64.41%, -23.0% to 37.1% and -42.9% to 62.4% (Fig. 6). Due to the large amounts of $NH_3$ emitted by fertilizers and livestock waste, the uncertainty of total $NH_3$ emissions is mainly caused by the uncertainties of these two sources. The uncertainty of fertilizer application was slightly greater than that of livestock waste. The emission factors, especially the corrected EF, were the largest contributors to the uncertainties of fertilizer application emissions. Additionally, it is clear that $NH_3$ emissions from other sources exhibited the largest uncertainty (-42.9% to 62.4%), mainly due to the high degree of uncertainty resulting from the many sub-sources, such as $-77.1\% \sim 96.9\%$ of the transportation sector and $-79.4\% \sim 122.7\%$ of the industrial sector. In comparison, the emissions from other sources were relatively small; hence, the large uncertainties of other sources did not have a significant impact on the uncertainty of total $NH_3$ emissions.

[Figure]

Fig. 6. Uncertainties of $NH_3$ emissions sourced from fertilizer application, livestock waste, and others."

3) The authors employed the WRF-Chem model for performing numerical simulations over eastern China and compared with model outputs driven by MEIC and IASI satellites. However, I could not find the detailed configurations of WRF-Chem simulations (initial and boundary conditions for WRF-Chem, physical parameterizations, chemical mechanisms, etc). I suggest the authors add a section in the supplementary providing the configurations of WRF-Chem.

Response: We sincerely thank the reviewer for this suggestion. We added one section (Section S1) to state the configurations of WRF-Chem model in the supplementary materials.

**"Section S1 WRF-Chem model configuration**

A fully coupled online Weather Research and Forecasting with Chemistry model (WRF-Chem v3.7) is used to evaluate the accuracy of different $NH_3$ emission inventories. The WRF-Chem model is designed to cover most parts of North, East, Central, and South China at the horizontal resolutions of 27 km (Fig.2). The vertical dimension is resolved by 46 full sigma levels, with 18 layers located in the bottom 2 km for finer resolution in the planetary boundary layer; the height of the first layer averaged over the analyzed region is about 30 m.

Meteorological initial and lateral boundary conditions used in the WRF-Chem model are taken from the NCEP (National Center for Environmental Prediction) (Final) Operational Global Analysis data with a spatial resolution of $1° × 1°$. The forecasts from the MOZART-4 global chemical transport model are processed to provide the chemical initial and boundary conditions for the WRF-Chem model (Emmons et al., 2010).

Air pollutants emissions (including $SO_2$, $NO_x$, CO, $CO_2$, NMVOC, BC, OC, $PM_{2.5}$, and $PM_{10}$)

of 2016 were obtained from Multi-resolution Emission Inventory for China (MEIC) (http://meicmodel.org/), with the horizontal resolution of 0.25° (Li et al., 2017). The emission rate of each species for each hour is based on Gao et al. (2015). The biogenic emissions are calculated online using the MEGANv2.04 (Model of Emission of Gases and Aerosol from Nature v2.04) model (Guenther et al., 2006). Biomass-burning emissions are obtained from the GFEDv3 (Global Fire Emissions Database v3) (Randerson et al., 2005). Dust emissions and sea salt emissions are calculated online using algorithms proposed by Shao (2004) and Gong et al. (1997), respectively.

The Carbon Bond Mechanism Z (CBM-Z) is selected as the gas-phase chemical mechanism (Zaveri and Peters, 1999), and the full 8-bin MOSAIC (Model for Simulating Aerosol Interactions and Chemistry) aerosol module with aqueous chemistry is used to simulate aerosol evolution (Zaveri et al., 2008). The photolysis rates are calculated by the Fast-J scheme (Wild et al., 2000). Aerosol radiation is simulated using RRTMG (Rapid Radiative Transfer Model for GCMs) for both shortwave (SW) and longwave (LW) radiation (Zhao et al., 2011). Other major physical parameterizations used in this study are listed in Table S5.

**Table S5 Parameterizations used in the WRF-Chem model**

| Options | WRF-Chem |
| --- | --- |
| Microphysics option | Purdue Lin Scheme |
| Longwave radiation option | RRTMG Scheme |
| Shortwave radiation option | RRTMG Scheme |
| Surface layer option | Revised MM5 Monin-Obukhov Scheme |
| Land surface option | Unified Noah land-surface model |
| Urban canopy model | Single-layer UCM Scheme |
| Boundary layer option | YSU Scheme |
| Cumulus option | Grell 3D ensemble Scheme |
| Photolysis scheme | Fast-J |
| Dust scheme | Shao_2004 |
| Chemistry option | CBMZ |
| Aerosol option | MOSAIC |

"

4) In the WRF-Chem simulations with NH3 emission inventory developed in this work and MEIC, I wonder whether other air pollutants emissions (such as SO2, NOx, VOCs) are both based upon MEIC. If it is, please clarify it.

Response: Yes, emissions of other air pollutants in 2016 (including $SO_2$, $NO_x$, CO, $CO_2$, NMVOC, BC, OC, $PM_{2.5}$, and $PM_{10}$) are both based upon MEIC in the WRF-Chem simulations. We added one sentence to clarify it in section 2.4.

"Emissions of other air pollutants in 2016 (including $SO_2$, $NO_x$, CO, $CO_2$, NMVOC, BC, OC, $PM_{2.5}$, and $PM_{10}$) were obtained from MEIC (http://meicmodel.org/), with a horizontal resolution of 0.25° (Li et al., 2017b)."

5) For readers not familiar with the China geography and the location of each province, providing a map with province names marked would be valuable.

Response: Thanks for your kindly suggestion. We marked the province names in the Fig. 2.

[Figure]

"

**Fig. 2. WRF-Chem simulation domain and sampling points (Ammonia Monitoring Network, China).**"

6) In Sec. 3.4, it seems the WRF-Chem model with optimized $NH_3$ emission inventory yields better performance in July while large bias still existed in January as compared with the MEIC driven WRF-Chem model. Could you be more specific on the reasons for the discrepancy between the simulated $NH_3$ concentrations and ambient measurements?

Response: We sincerely thank the reviewer for this suggestion. As suggested, we added one paragraph in Section 3.4.3 to explain the possible reasons for the discrepancy between the simulated $NH_3$ concentrations and ambient measurements in January and October.

"In general, our inventory exhibited better spatial accuracy than other inventories that utilize WRF-Chem and observations, although January and October showed a relatively large bias between measured and simulated data. The $NH_3$ concentrations were low in these two months. We found that the accuracy of the simulation increased as the concentration of atmospheric $NH_3$ increased. The mean values of IASI $NH_3$ VCDs in the simulation domain in January and October were $7.2\times10^{15}$ and $8.7\times10^{15}$ molec/cm$^2$, respectively, significantly lower than the mean values of $14.9\times10^{15}$ and $19.4\times10^{15}$ molec/cm$^2$ in April and July. In January and October, the relatively low $NH_3$ emissions (32.5% and 54.6% of July emissions) combined with the short lifetime of $NH_3$ and uncertainties in gaseous $NH_3$ and aerosol $NH_4^+$ partitioning pose a challenge to the chemical mechanisms of the WRF-Chem model, making it more difficult for the model to fully capture the heterogeneity of $NH_3$ concentration. Besides, the relatively large uncertainty of the IASI VCDs could also contribute to

inconsistency between simulated and observed concentrations (Van Damme et al., 2017; Chen et al., 2020). Zhang et al. (2018) and Zhao et al. (2020) obtained similar results which indicated that the correlations between measured and simulated $NH_3$ data were lower in January and October than during the other two months."

7) I think Figure 6(a) is misleading with the erroneous higher concentrations of NH3 in January than July due to the different scales in colormap. Please use the same color scale for both January and July. Otherwise, it is misleading to the readers.

Response: According to your advice, we used the same color scale for both January, April and July in Fig.7 (the original Fig.6).

8) Conclusion: This section would be stronger if a discussion on future work that the community should consider toward improving the representation of ammonia emissions in chemical transport models (For example, investigating the impacts of optimized ammonia emission inventory on simulating $PM_{2.5}$ concentrations and aerosol pH changes).

Response: We sincerely thanks for your constructive suggestion. As suggested, we illustrated the potential applications of our improved ammonia emission inventory in the Conclusion.

"We believe that the improved $NH_3$ emission inventory can be used in future research, to simulate atmospheric aerosol formation, investigate the influence of $NH_3$ emission on $PM_{2.5}$ mass burden and aerosol pH changes, develop targeted $NH_3$ reduction strategies to further improve air quality, and explore the atmospheric N cycle process."

Specific comments:

1) Line 17: "differences"-->"variability". "observed" -->"identified".

Response: We have corrected it.

2) Line 84: "huge"-->"considerable".

Response: Corrected.

3) Line 215: "30 samples" should be "30 sampling stations".

Response: Corrected.

4) Line 223: "represented"-->"contributed".

Response: Corrected.

5) Line 223: The 13.1 Tg is derived from Kong et al., (2019). Thus, there is a lack of citation for Kong et al., (2019).

Response: Thanks for pointing it out, we added this citation.

6) Table 2: I suggest the authors include the ammonia emissions quantified in this work in Table 2 as well.

Response: We added the $NH_3$ emission value of this study in Table 2.

7) Line 277: Please further clarify the reason why located in north China is relevant to elevated ammonia emissions.

Response: We added more details to explain the reasons for the high NH$_3$ emission density in the North China Plain.

"These provinces are all concentrated in the North China Plain (NCP, which includes the above six provinces and Beijing), an area that features well-developed crop farming and animal husbandry. The NCP contains a large amount of high-quality arable land, and the farms in this area produce 34.47% of China's major farm products (NBS, 2017c). The fertilization of crops emits large amounts of NH$_3$. The soil in this area is alkalescent (the average pH value is 7.15), which further increases the NH$_3$ volatilization. Additionally, the NCP produces as much as 28.16 million tons of pork and beef per year, accounting for 32.5% of the total (NBS, 2017c), contributing to higher NH$_3$ emissions in the North China Plain."

8) Line 313: Please rephrase this sentence for read.

Response: We have revised this sentence as follows:

"Unlike some pollutants (such as PM$_{2.5}$ and BC) which exhibit higher emissions in winter, NH$_3$ emissions tended to be high in summer and low in winter."

9) Line 324: Delete "it".

Response: Deleted.

10) Line 349: Delete "Thus".

Response: Deleted.

11) Line 359: The spatial resolution for MEIC is 0.25°×0.25°.

Response: Corrected.

12) Figure S2: Small typo for the caption of subplot for Others.

Response: Corrected.

13) Some figures are hard to read due to the small font size. Please consider improving them.

Response: Thanks for your critical comment. As suggested, we enlarged the font size and improved the resolution of Fig. 4, Fig. 7 and Fig. 8.

14) The manuscript is hard to follow in several places but that could be addressed with thorough language editing.

Response: The manuscript has been edited via professional editing service to reduce grammatical errors.

---

## Author Comment (AC2)

General comments:

This manuscript presented a new bottom-up estimate of ammonia emissions in China that emphasized some improvements on the estimates of fertilizer application-induced ammonia volatilization. The authors compared their Chinese ammonia emission inventory with a commonly-used emission estimate (MEIC), and evaluated the resulting model simulations using the WRF-Chem air quality model with surface and satellite ammonia measurements.

The manuscript is in general well conducted and organized, and it meets the scope of ACP. I have some comments below that suggest the authors better clarify the improvements of this ammonia emission inventory relative to previous studies. I think these comments should be addressed before considering publish.

Response: We really appreciate the valuable and constructive comments from the reviewer. We have made changes to both the main text and the supplemental information. Detailed responses are shown below.

Specific comments:

1) Page 5: It appears that most of the methods and datasets applied in this NH3 emission inventory have followed previous studies (Huang et al., 2012; Kang et al., 2016; Zhang et al., 2018), except here as described in this section the improvements on fertilizer application-related NH3 emission estimate. It is important to better identify the fertilizer application timing and proportion throughout the planting growing season. The key improvement in this ammonia emission inventory is depicted in Figure 1 that accounts for the spatial differences in fertilizer application timings for the three main crops (maize, wheat, and rice). From Figure 1, it is not clear what these fertilizer application dates are for each crop and each region. I think this information shall be presented, e.g., as a table in the manuscript or in the Supplement.

In addition, how the fertilizer amounts are distributed in each month according to the application timing needs some description in the text. Did you use a uniform fertilizer application rate for each crop in each month and region?

Response: We sincerely thank you for providing this constructive comment. As suggested, we added one Table (please see the Table S1 in the Supplementary materials) to show the basal and topdressing fertilization dates for different crops in different provinces.

In addition, we also reorganized the Section 2.1 and added one equation (equation (2)) to show how the N fertilizer amounts are distributed in different months.

"The respective amounts of each N fertilizer, applied in different months to each crop category, can be calculated using the following formula:

$$A_{i,j} = A_{total} \times P_j \times x_{i,j} \tag{2}$$

where i indicates the month. $A_{total}$ indicates the total amount of each N fertilizer (ABC, DAP, NPK, urea, and other) applied to the different crop types in each province in 2016, which were calculated as a product of the planted cropland area and the fertilizer application rate per unit area of cropland,

based on data from MARA (2017) and NDRC (2017). Over the whole growth period, wheat, maize, and rice generally need three applications of N fertilizer ($j$, namely the basal dressing, first topdressing, and second topdressing). $P_j$ represents the proportion of the total annual fertilization amount applied in the $j$-th dressing. The variation in $P_j$ across the different regions was also considered, based on farmer survey data (Wang et al., 2008) and other studies in the literature (Zhang et al., 2009; Zhang and Zhang, 2012). $x_{i,j}$ represents the probability of the $j$-th dressing in month $i$, and is calculated as the proportion of days in the $i$-th month during the window period of $j$-th dressing. Table S1 lists the basal and topdressing fertilization dates for different crops in different provinces.

An example: in 2016, winter wheat was sown and basal fertilizer was applied in late September to mid-October. So, the dates of basal dressing in Hebei province span two months; $x$=1/3 in September, while $x$=2/3 in October. Similarly, the other two top dressings were applied at the jointing and booting stages, i.e., in late March to early April and late April to early May, respectively. According to the proportion of basal dressing and top dressing for wheat in Hebei, we identified the proportion of total fertilizer applied in each month (basal dressing: 0.2, September; 0.4, October; top dressing: 0.1, March; 0.2, April; and 0.1 in May).”

2) Page 9, Section 2.4: The results of Monte Carlo calculation were not presented in the manuscript. What are the uncertainties and probability distributions of ammonia emissions from fertilizer and livestock?

Response: We sincerely thank you for this critical comment. We added one sub-section (Section 3.4.1) to present the uncertainty assessment of $NH_3$ emission based on the Monte Carlo simulation.

**“3.4.1 Uncertainty**
Uncertainty in the estimated $NH_3$ emissions results from both the activity level and EF input data. We ran 10000 Monte Carlo simulations to estimate the range of $NH_3$ emissions from each source with a 95% confidence interval. The estimated total $NH_3$ emission range was 10.5-16.0 Tg. The 95% confidence intervals of fertilizer application, livestock waste, and others ranged from -20.5% to 64.41%, -23.0% to 37.1% and -42.9% to 62.4% (Fig. 6). Due to the large amounts of $NH_3$ emitted by fertilizers and livestock waste, the uncertainty of total $NH_3$ emissions is mainly caused by the uncertainties of these two sources. The uncertainty of fertilizer application was slightly greater than that of livestock waste. The emission factors, especially the corrected EF, were the largest contributors to the uncertainties of fertilizer application emissions. Additionally, it is clear that $NH_3$ emissions from other sources exhibited the largest uncertainty (-42.9% to 62.4%), mainly due to the high degree of uncertainty resulting from the many sub-sources, such as $-77.1\% \sim 96.9\%$ of the transportation sector and $-79.4\% \sim 122.7\%$ of the industrial sector. In comparison, the emissions from other sources were relatively small; hence, the large uncertainties of other sources did not have a significant impact on the uncertainty of total $NH_3$ emissions.

[Figure]

Fig. 6. Uncertainties of NH$_3$ emissions sourced from fertilizer application, livestock waste, and others."

3) Page 17, Section 3.4: The study pointed out that Chinese ammonia emissions are high in summer and low in winter, and confirmed the results using WRF-Chem model simulations and ammonia measurements. This result did not seem to be much improved compared with previous estimates that all suggested higher ammonia emissions in summer than winter. I think that only analyzing the two months (January and July) could not provide sufficient information on the improvements of the ammonia emission inventory. The new fertilizer-induced ammonia emissions (Figure 5) also high values in April and October? Can you also evaluate improvements in these two months? This will provide valuable information to understand ammonia in spring and fall seasons.

Response: We appreciate the reviewer for providing this comment. As suggested, we further evaluated the spatial accuracy of the NH$_3$ emission inventory established in this study and the MEIC inventory in April and October 2016. It should be note that for October, we only used the IASI satellite observations to evaluate the spatial accuracy of the two inventories, because ground observation data were not available for October. The detailed accuracy evaluation was described in the Section 3.4.3.

4) Page 18, Fig. 6: The color scale for the left panel of Fig 6 could be misleading. It shows that at many sites, the model results are too high over the North China plain, however, the model results (contours) and measurements (dots) have different color scales. Suggest put them on the same color scale.

Response: Thanks for your valuable comment. According to your advice, we used the same color scale in Fig.7 (the original Fig.6).

5) Page10, Line 220: Here "9.29-15.54 Gg" should be in unit of "Tg".

Response: We are sorry for this mistake. we have corrected it.

6) Page 11, Line 245: "replacing complex fertilizers with ABC might reduce fertilizer type-related NH3 emissions". Should here be "replacing ABC with complex fertilizer"?

Response: we have corrected it.

7) Page 17, Line 337: "R$^2$=0.85" should be 0.84 as shown from Table 4. Was the R$^2$ value calculated by integrating January and July measurements?

Response: We have corrected it. The $R^2$ value was obtained by fitting the 2016 monthly values between IASI satellite observations and the NH$_3$ emissions from two inventories (MEIC and our study). To avoid this confusion, we revised the relevant sentence and added a Note in Table 4.

"We verified the accuracy of the monthly NH3 emission trends obtained in this study by determining the correlation between the monthly IASI satellite observations and the monthly emissions of two inventories (MEIC and our study) in 2016."

"Table 4. Comparison of the monthly NH$_3$ emission trends corresponding to IASI satellite observations with the two inventories.

| Region | $R^2$ | | July/Jan ratio | | |
|--------|-------|------|------|-----|------|
| | IASI vs. OUR | IASI vs. MEIC | IASI | OUR | MEIC |
| … | … | … | … | … | … |

Note: $R^2$ value was obtained by fitting the 2016 monthly values between IASI satellite observations and the NH3 emissions from two inventories (MEIC and our study)."

---

## Author Response (AR2)

**Response to Editor:**

Thank you for your careful consideration of the referee comments. I find that the revisions have largely addressed the referee comments and am happy to accept the paper for publication following attention to the following comments. Line numbers refer to the track changes version of the manuscript.

Response: We appreciate that the editor and reviewers recognize our efforts and thank you for your thoughtful suggestions and insights, which have helped improve this manuscript substantially. The detailed responses are listed as follows.

1) lines 29-30: To my understanding, $NH_3$ reacts with oxidized organics within the condensed phase under atmospheric conditions, but I am unaware of atmospherically relevant reactions with alkanes. The cited manuscript does not support that reaction either. Please clarify.

Response: We agree with the comment and apologize for this mistake. We revised this sentence as follows:

"For example, in China, the contribution of agriculture-related NH3 emissions to SIA and $PM_{2.5}$ is 29% and 16%, respectively (Han et al., 2020). This fine particle formation has led to substantial health and economic costs (Paulot and Jacob, 2014)."

2) Line 107: These reports need to be better documented/referenced. Depending on the number, citation in the paper may not be appropriate, but the SI or an archived document may be sufficient.

Response: We sincerely thank you for providing this valuable comments. As suggested, we added one section (Section S1) and a table (Table S5) to state the main data sources of the fertilization application timing and frequency for the three main crops: rice, maize, and wheat. Most of the collected reports/websites were published by the national or provincial governments in 2016.

3) Please increase the text size in Fig. 3.

Response: As suggested, we increased the text size in Fig.3.

[Figure]

Fig. 3. Contributions of different sources (%) to $NH_3$ emissions in mainland China (2016)."

4) Table 4: Why are some numbers in bold?

Response: Thanks for pointing it out. The $R^2$ values in bold represent improved monthly $NH_3$ emission trend in our inventory compared to that of the MEIC inventory in the corresponding regions. We also explained it in the Note.

"Note: $R^2$ value was obtained by fitting the 2016 monthly values between IASI satellite observations and the $NH_3$ emissions from two inventories (MEIC and our study). The $R^2$ values in bold represent improved monthly $NH_3$ emission trend in our inventory compared to that of the MEIC inventory in the corresponding regions."

5a) Sect 3.4.3: Please comment on the y-intercepts of the trend lines between the simulated measurements and ground based concentrations as well. There appears to be a significant positive intercept suggesting overestimation despite the slope near one. To me this suggests that the variation in spatial distribution is perhaps better than the absolute concentrations. The MEIC comparison also shows a positive y-intercept so it is not clear if it is a true underestimation of emissions (line 396).

Response: We sincerely thank you for providing this critical comment. We believe that positive intercepts mainly resulted from simulation results that overestimated the $NH_3$ concentrations at relatively low values (Fig. 7(b)) (e.g., the simulation for January). However, the MEIC inventory may have underestimated the $NH_3$ concentration in areas with a high emission density (e.g., the simulation for the North China Plain in July). We changed the relevant sentences as follows:

"The y-intercepts of the trend lines between the ground observations and the simulated measurements using MEIC and our inventory were respectively 5.0 and 4.0. Positive intercepts mainly resulted from simulation results that overestimated the $NH_3$ concentrations at relatively low values (Fig. 7(b)) (e.g., the simulation for January). However, the MEIC inventory may have underestimated the $NH_3$ concentration in areas with a high emission density (e.g., the simulation for the North China Plain in July)."

5b) Sect 3.4.3: Lines 392-395: I assume that the $R^2$ and slopes reported are the comparison when all months are included, but this is not explicitly stated. Are there differences if months are compared individually? Please clarify.

Response: Thanks for your valuable comment. In the revised version, we compared the $R^2$ and slopes obtained by fitting the ground-based observations with $NH_3$ concentrations simulated using MEIC and our inventory of January, April and July, respectively. We found similar results for the spatial accuracy of our inventory, which were better than that of the MEIC.

"For each month, we found similar results for the spatial accuracy of our inventory, which were better than that of the MEIC. The $R^2$ values and slopes obtained by fitting the ground-based observations with the $NH_3$ concentrations simulated via the MEIC yielded an $R^2$ of 0.18 and slope of 0.54 in January, $R^2$ of 0.01 and slope of 0.11 in April, and $R^2$ of 0.21 and slope of 0.28 in July, which were significantly lower than those obtained by fitting the simulated $NH_3$ concentrations using our inventory and the ground-based observations ($R^2$ of 0.27 and slope of 0.68 in January, $R^2$ of 0.23 and slope of 0.70 in April, and $R^2$ of 0.53 and slope of 0.87 in July)."

5c) Figure 7b: I find it difficult to tell the different months apart in this graph and this makes it challenging to assess differences between the months. I suggest considering different symbols rather than different sizes or adding plots to the SI that shows the months individually.

Response: Thanks for your comment. As suggested, we used different symbols to represent the ammonia concentrations in different months.

[Figure]

Fig. 7. (a) Spatial distribution of NH₃ concentrations in 2016, from this study and the MEIC inventory. (b) Correlation between simulated NH₃ concentrations, from different emission inventories, and ground observations obtained from AMoN-China."

6) Fig 8: Please use the same color scale for all plots or explicitly call the reader's attention to the fact that they vary.

Response: Thanks for your kindly comment. The values of NH₃ VCDs in different months vary greatly. In order to better present the spatial distribution of NH₃ VCDs, different color scales are used for different months. In the figure caption, we added one sentences to emphasize the difference in color scale.

"Fig. 8. The spatial distribution of IASI NH₃ VCDs and NH₃ VCDs from WRF-Chem based on the two inventories in January, April, July, and October (2016). Different color scales represent different months to indicate the spatial distribution of the NH₃ VCDs."

7) Figs. S1 & S2: Please increase resolution and font size. The color scale values are not legible.

Response: We revised the Fig. S1 and Fig. S2 based on your suggestion.

[Figure]

"

Fig. S1. Spatial distribution of single-season rice, middle rice and early/late rice.

[Figure]

Fig. S2. Geographical distribution of NH$_3$ emission from fertilizer application, livestock wastes, and others in mainland China (2016)."

8) Fig. S3: please explicitly draw the reader's attention to the fact that the range on the plots varies.

Response: In the figure caption, we added one sentences to state the range on the plots varies.

"Fig. S3. Comparison between IASI-based VCDs and simulated NH$_3$ VCDs obtained in this study and MEIC, for January, April, July, and October. The range of the axes on the scatter plots for the different months is not the same."

9) I urge the authors to consider, but I do not require, depositing at least some parts of the data in a public data repository to foster accessibility and citation. Please see this website for the data policy: https://www.atmospheric-chemistry-and-physics.net/policies/data_policy.html

Response: Thanks for your comment. We have made our gridded ammonia emission inventory publicly available based on your suggestion. The gridded ammonia emission inventory is archived on Zenodo (https://doi.org/10.5281/zenodo.5516929). The relevant sentences in the *Data availability* were also updated.